# Efficient Face Super-Resolution via Wavelet-based Feature Enhancement Network

## ABSTRACT

Face super-resolution aims to reconstruct a high-resolution face image from a low-resolution face image. Previous methods typically employ an encoder-decoder structure to extract facial structural features, where the direct downsampling inevitably introduces distortions, especially to high-frequency features such as edges. To address this issue, we propose a wavelet-based feature enhancement network, which mitigates feature distortion by losslessly decomposing the input facial feature into high-frequency and low-frequency components using the wavelet transform and processing them separately. To improve the efficiency of facial feature extraction, a full domain Transformer is further proposed to enhance local, regional, and global low-frequency facial features. Such designs allow our method to perform better without stacking many network modules as previous methods did. Extensive experiments demonstrate that our method effectively balances performance, model size, and inference speed. All code and data will be released upon acceptance.

## CCS CONCEPTS

• **Computing methodologies** → **Reconstruction**; **Computational photography**; **Image processing**.

## KEYWORDS

Face super-resolution, Efficient, Wavelet transform, Full Domain Transformer.

## 1 INTRODUCTION

Face super-resolution (FSR), also known as face hallucination, aims to convert a low-resolution (LR) face image into a high-resolution (HR) face image. Different from image super-resolution, FSR focuses on reconstructing essential structural information about the face, including facial contours and the shape of facial components. This paper aims to propose a high-fidelity FSR method while maintaining efficiency in model size and inference speed, as depicted in Fig. 1.

Existing FSR methods [2, 4, 15] typically apply an encoder-decoder structure, which is due to this structure facilitates the model to grasp the overall facial structure during the encoder stage with a larger receptive field and enhances facial details during the decoder stage. Specifically, the encoder initially downsamples the LR input, extracting facial features at various scales. Subsequently, the decoder progressively upsamples the features from the encoder

Permission to make digital or hard copies of all or part of this work for personal or classroom use is granted without fee provided that copies are not made or distributed for profit or commercial advantage and that copies bear this notice and the full citation on the first page. Copyrights for components of this work owned by others than the author(s) must be honored. Abstracting with credit is permitted. To copy otherwise, or republish, to post on servers or to redistribute to lists, requires prior specific permission and/or a fee. Request permissions from permissions@acm.org.

*MM'24, October 28 - November 1, 2024, Melbourne, Australia.*

© 2018 Copyright held by the owner/author(s). Publication rights licensed to ACM.
ACM ISBN 978-1-4503-XXXX-X/18/06
https://doi.org/XXXXXXX.XXXXXXX

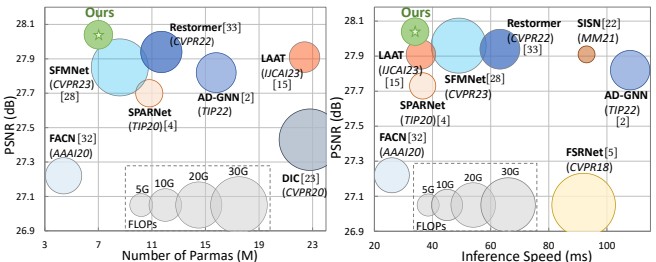

(a) PSNR, FLOPs and Params Tradeoffs.  (b) PSNR, FLOPs and Speed Tradeoffs.

**Figure 1: Efficiency trade-offs between ours and state-of-the-art methods on the CelebA [21] test set. Our method achieves a better balance in terms of performance (PSNR), model size (number of Params and FLOPs), and inference speed.**

output, refines details, and ultimately generates the HR output. However, previous methods neglect the impact of the chosen downsampling technique on the restoration outcomes. For example, some methods [4, 7] employ bicubic interpolation or strided convolution for downsampling. These operations reduce the number of image pixels, potentially leading to the loss of critical facial details essential for reconstruction. As depicted in Fig. 2 (b) and (c), bicubic interpolation and strided convolution lead to a significant loss of texture in the overall facial structure, resulting in distortions in the subsequent reconstruction of the face profile. Another example from [12] utilizes downsampling through avgpooling operations. As illustrated in Fig. 2 (d), this results in the aliasing artifact of high and low-frequency facial features. This phenomenon is particularly evident in the eye features and significantly hampers the accurate representation of facial components.

To address the above problem, we propose to utilize discrete wavelet transform to enhance the features. Specifically, following the Nyquist sampling theorem, the standard downsampling process involves a low-pass filter followed by downsampling to prevent frequency domain aliasing. Discrete wavelet transform [24] can simulate standard downsampling by decomposing the input image $I \in \mathbb{R}^{H \times W}$ into four components at different frequencies, which consist of one low-frequency component $I_{LL} \in \mathbb{R}^{\frac{H}{2} \times \frac{W}{2}}$ and three high-frequency components $\{I_{LH}, I_{HL}, I_{HH}\} \in \mathbb{R}^{\frac{H}{2} \times \frac{W}{2}}$. The low-frequency component $I_{LL}$ can be approximated as the result obtained after low-pass filtering followed by downsampling. Simultaneously, the acquired high-frequency components can still be fused with low-frequency features to enhance information, such as face edges, to alleviate the problem of facial structure loss caused by downsampling. As shown in Fig. 2 (e), employing wavelet feature decomposition and fusion results in significantly clearer facial contours, and there is no occurrence of frequency domain aliasing. This result demonstrates the strategy's effectiveness involving using wavelet transform to decompose features for downsampling

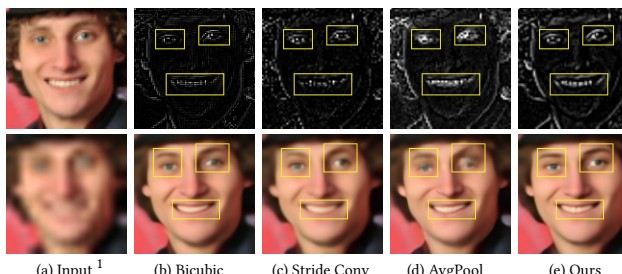

(a) Input [1]    (b) Bicubic    (c) Stride Conv    (d) AvgPool    (e) Ours

**Figure 2: Feature maps (first line) and FSR results (second line) with various downsampling methods: bicubic interpolation, stride convolution, average pooling, and our wavelet feature downsample. The loss of high-frequency features is pronounced in (a) and (b), while frequency-domain feature aliasing is prominent in (c). Ours is effective in avoiding the above feature loss or frequency-domain aliasing.**

and further utilizing the decomposed high-frequency features to enhance the face profile. Inspired by this observation, we introduce wavelet feature downsample (WFD) and wavelet feature upgrade (WFU). WFD aims to minimize distortion of crucial facial structures during downsampling in the encoder phase. WFU aims to enhance facial contour by utilizing extra features obtained through wavelet decomposition in the decoder phase.

To better enhance the low-frequency facial information obtained after wavelet transform decomposition, we introduce a full-domain Transformer (FDT). Specifically, as the low-frequency information encapsulates the main features of an image, extracting comprehensive low-frequency information is crucial. Despite the Transformer demonstrating efficacy in handling low-frequency information, the Transformer utilized in existing FSR methods struggles to effectively concentrate on local features (*e.g.*, skin details), regional features (*e.g.*, components like eyes, noses), and global features (*e.g.*, the overall face profile of the face). To address this problem, FDT is proposed to explore diverse receptive fields and uncover deeper correlations within facial features to extract more comprehensive information to enhance facial details.

By utilizing WFD and WFU to alleviate facial feature distortion and employing FDT for comprehensive extraction of facial features, our wavelet-based feature enhancement network (WFEN) achieves robust performance without the need for excessive network module stacking like previous methods. Results as Fig. 1, our WFEN demonstrates outstanding efficiency compared to state-of-the-art methods. In summary, the contributions of this paper are as follows:

- We propose WFD and WFU modules utilizing wavelet transform to minimize the distortion of facial features and enhance face contour in the encoder-decoder structure.
- We propose an FDT module that extends interactions to the local, regional, and global levels, providing our model with richer facial receptive field information.
- We propose a WFEN, which is more efficient than state-of-the-art methods and achieves a better balance in performance, model size, and inference speed.

---

[1]The clearer input for the first line is to make the feature map easily observable and the contrast pronounced.

**Table 1: Comparison of encoder-decoder-based network design in existing methods.**

| Methods | Wavelet-based | What to focus on? |
|---|---|---|
| SPARNet [4] | No | Spatial attention. |
| Restormer [33] | No | Channel-based self-attention. |
| LAAT [15] | Yes | Feature fusion from coarse to fine. |
| SFMNet [28] | No | Utilizing Fourier domain feature. |
| **Ours** | Yes | Mitigating feature corruption in downsample. |

## 2 RELATED WORK

Since our method enhances FSR performance through the application of wavelet theory, we present recent advances in FSR and discuss the utilization of wavelet transform in super-resolution. The difference with the main related methods can be seen in Table 1.

### 2.1 Face Super-resolution

With the advancements in deep learning, numerous neural networks [18] for FSR have been proposed to enhance performance. Due to the highly structured nature of the human face, one category of methods aims to leverage facial priors, such as facial landmarks [23], facial parsing maps [5], facial attribute [32], 3D facial shapes [10], *etc.*, to assist in the restoration process. However, incorporating the face prior estimation module into the network will unavoidably introduce an additional computational burden. Moreover, accurately estimating facial geometric priors becomes highly challenging when dealing with very low-resolution face images. Inaccurate estimation of face priors frequently results in a substantial distortion of the restoration outcomes, which limits the development of prior-based methods in the field of FSR.

Consequently, attention-based FSR methods have gained prominence. RAAN [31] utilizes channel attention to extract face shape features, significantly enhancing the model's expressive power. SPARNet [4] introduces spatial attention, allowing it to capture facial structural features efficiently. SISN [22] separately explores facial structural information and facial texture details through the external-internal separation of attention. AD-GNN [2] leverages a series of spatial attention to explore feature relationship modeling and complement facial details. To simulate long-distance modeling, FaceFormer [29] leverages Transformer's capabilities for long-distance modeling to capture global facial information. LAAT [15] further enhances fine-grained regions of features by introducing a self-refinement mechanism into the Transformer. CTCNet [7] and SCTANet [3] integrate spatial attention and self-attention within the model to effectively leverage local and global information. SFM-Net [28] employs frequency domain branching and spatial branching to extract global and local correlations, respectively. However, none of them consider the detrimental effect of downsampling in the encoder-decoder on reconstruction. Moreover, unlike these methods that focus only on local or global facial features, our method simultaneously focuses on local, regional, and global facial features.

### 2.2 Wavelet Transform-based Methods

Recently, wavelet theory has gained prominence in super-resolution. DWSR [9] employs CNN representations on low-resolution wavelet

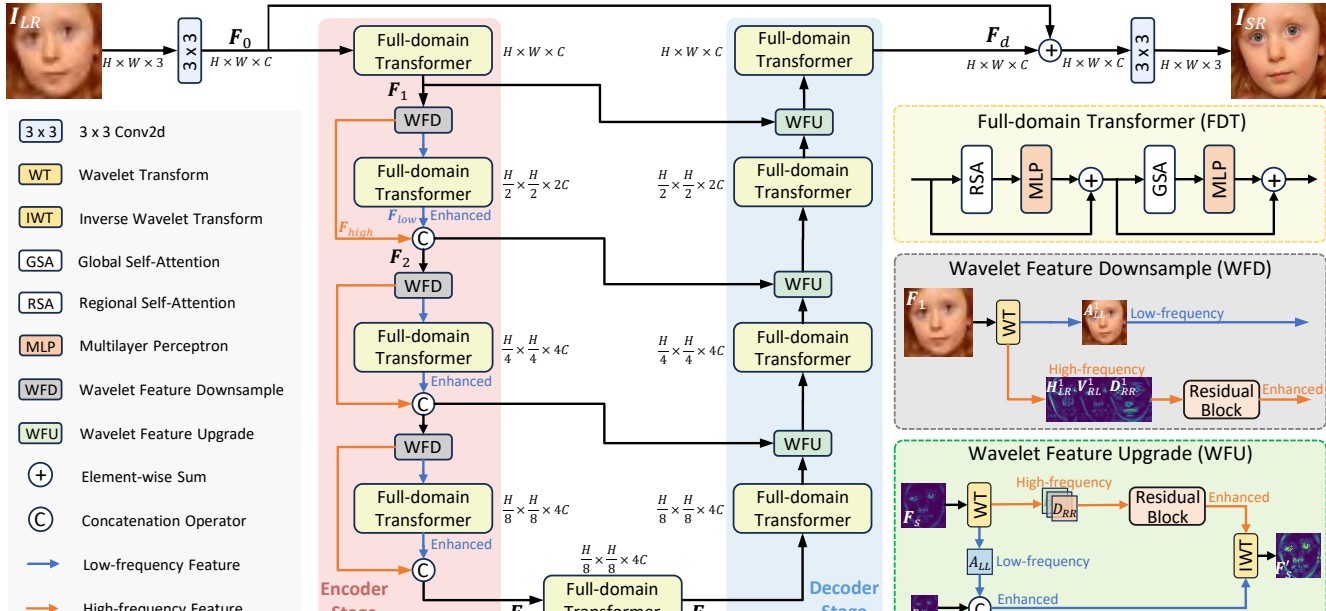

**Figure 3: Overview of our method, where cascading WFD and WFU constitute the wavelet-based encoder-decoder structure.**

subbands to recover missing details. Wavelet-SRNet [11] reconstructs a face image from a sequence of wavelet coefficients of the HR corresponding to the LR learned by the network. SRCliqueNet [36] explores relationships between wavelet transform subbands to aid the reconstruction process. JWSGN [37] employs wavelet transform to reconstruct the frequency domain details of images. WTRN [20] reconstructs the texture by computing the correlation of the wavelet-transformed subbands with the reference image. FOF [16] considers data characteristics in the frequency domain through wavelet transforms, thereby enhancing the efficiency of the network. LAAT [15] employs a wavelet fusion module to combine shallow structures and deep details to recover realistic images in the frequency domain. Unlike these methods, we focus on utilizing wavelet transform to decompose the high and low-frequency components for lossless downsampling, which in turn reduces the feature corruption of downsampling in encoder-decoder structure.

## 3 PROPOSED METHOD

### 3.1 Overview

As shown in Fig. 3, from a given degraded face image $I_{LR} \in \mathbb{R}^{H \times W \times 3}$, we aim to reconstruct a clean face $I_{SR} \in \mathbb{R}^{H \times W \times 3}$ by employing a wavelet-based encoder-decoder structure integrated with residual block and our full-domain transformer. The wavelet-based encoder-decoder structure encompasses our wavelet feature downsample in the encoder stage and wavelet feature upgrade in the decoder stage for downsampling and upsampling.

Specifically, our method initially extracts shallow facial features $F_0 \in \mathbb{R}^{H \times W \times C}$ from $I_{LR}$, where $H \times W$ denotes the spatial resolution, and $C$ denotes the number of channels. Subsequently, $F_0$ undergoes hierarchical level-by-level processing through wavelet

feature downsample, gradually transforming the $F_0$ into a low-resolution latent representation $F_1 \in \mathbb{R}^{\frac{H}{8} \times \frac{W}{8} \times 4C}$. At each level, the low-frequency part of the transform is fed to our full-domain Transformer, while the high-frequency part is fed to the residual block. During the bottleneck stage, situated between the encoder and decoder stages, a sequence of full-domain Transformers is employed to refine $F_1$ to obtain $F_e \in \mathbb{R}^{\frac{H}{8} \times \frac{W}{8} \times 4C}$. Then, we incorporate wavelet feature upgrade before each decoding level, which effectively performs cross-scale feature fusion to obtain accurate depth features $F_d \in \mathbb{R}^{H \times W \times C}$. Finally, the output $F_d$ from the decoder stage recovers a clean face image $I_{SR}$ after residual concatenation and dimensionality reduction. In the following subsections, we provide a detailed description of the core modules we have constructed.

### 3.2 Wavelet-based Encoder-Decoder Structure

As depicted in Fig. 3, the central components of the wavelet-based encoder-decoder structure consist of a series of wavelet feature downsamples in the encoder and a series of wavelet feature upgrades in the decoder. They are tasked with progressively downsampling and upsampling, forming the main structure of our network.

*Wavelet Feature Downsample (WFD).* During the encoder process, downsampling is typically employed to decrease the size of the feature map. However, as mentioned above, existing methods overlook the irreversible distortion caused by downsampling, resulting in unclear edges in the FSR results. It occurs because traditional downsampling operations, which decrease resolution by merging neighboring pixels, can result in feature distortion, particularly in regions with significant gradient changes, due to the reduction in sampling points. In this context, as shown in Fig. 3, we introduce a

**Figure 4: Architecture of our proposed full-domain Transformer, which can focus on local, regional, and global facial features.**

WFD using wavelet transform to alleviate this phenomenon. Additional information about wavelet transform can be found in our *supplementary materials*.

For input facial feature $F_1 \in \mathbb{R}^{H \times W \times C}$, we initially apply a wavelet transform $\mathcal{WT}$, allowing us to decompose $F_1$ into four sub-wavelet bands: low-pass feature $A_{LL}^1$, and high-frequency facial in horizontal, vertical, and diagonal directions $H_{LR}^1$, $V_{RL}^1$, and $D_{RR}^1$:

$$\{A_{LL}^1, H_{LR}^1, V_{RL}^1, D_{RR}^1\} = \mathcal{WT}(F_1), \tag{1}$$

where $\{A_{LR}^1, H_{LR}^1, V_{RL}^1, D_{RR}^1\} \in \mathbb{R}^{\frac{H}{2} \times \frac{H}{2} \times C}$. As the low-frequency part predominantly carries essential information in the image, we focus on low-frequency face details on the main path while paying attention to high-frequency face textures on the residual path:

$$F_{low}, F_{high} = \mathcal{T}\left(A_{LL}^1\right), \mathcal{R}\left(H_{LR}^1, V_{RL}^1, D_{RR}^1\right), \tag{2}$$

where $F_{\text{low}}$ denotes the enhanced low-frequency features, $F_{\text{high}}$ denotes the enhanced high-frequency features, $\mathcal{T}$ denotes our full-domain Transformer, and $\mathcal{R}$ denotes residual block. We opt for different structures to extract high and low-frequency features because prior researches [14, 26] indicate that Transformer is more sensitive to low-frequency features, while CNN is more sensitive to high-frequency features. Next, the full downsampled enhanced feature $F_2 \in \mathbb{R}^{\frac{H}{2} \times \frac{H}{2} \times 2C}$ is obtained by fusing $F_{\text{low}}$ and $F_{\text{high}}$. With this thoughtful design, our model can enhance efficiency in handling both high and low-frequency facial features.

*Wavelet Feature Upgrade (WFU).* To obtain more details, several methods [2, 7, 15] propose using residual concatenation to enable the decoder to leverage information from the encoder. As the resolution of features at different scales differs, upsampling is employed to align them to the same resolution before feature fusion. Nevertheless, direct fusion operation is not optimal as it may introduce some degree of high and low-frequency aliasing. To better fuse features from the encoder, we leverage the wavelet transform for image scale transformations, developing a WFU to effectively utilize features from different scales in the decoder to enhance facial details by fusing high and low-frequency features separately.

Specifically, as shown in Fig. 3, for larger scale feature $F_s \in \mathbb{R}^{\frac{H}{4} \times \frac{W}{4} \times 4C}$ from the encoder and smaller scale feature $F_{s+1} \in \mathbb{R}^{\frac{H}{8} \times \frac{W}{8} \times 4C}$ from the decoder, we initially apply the wavelet transform to the larger scale feature $F_s$, resulting in four wavelet subbands of same scale as $\mathbb{R}^{\frac{H}{8} \times \frac{W}{8} \times 4C}$:

$$\{A_{LL}^s, H_{LR}^s, V_{RL}^s, D_{RR}^s\} = \mathcal{WT}(F_s), \tag{3}$$

where $A_{LL}^s$ represents the low-frequency part of $F_s$ and $H_{LR}^s$, $V_{RL}^s$, $D_{RR}^s$ represent the three high-frequency parts of $F_s$. Considering that small-scale feature $F_{s+1}$ is presumed to contain low-frequency information predominantly, we combine $A_{LL}^s$ with it as the enhanced low-frequency subband. Simultaneously, we employ a residual block to strengthen the high-frequency components of the image, and ultimately, output $F_s' \in \mathbb{R}^{\frac{H}{4} \times \frac{W}{4} \times 4C}$ can be obtained through the inverse wavelet transform:

$$F_s' = \mathcal{IWT}\left(C\left(A_{LL}^s, F_{s+1}\right), \mathcal{R}\left(H_{LR}^s, V_{RL}^s, D_{RR}^s\right)\right), \tag{4}$$

where $\mathcal{IWT}$ denotes the inverse wavelet transform, $C$ denotes the concatenation, and $\mathcal{R}$ denotes the standard residual block.

## 3.3 Full-domain Transformer

As analyzed above, the main path of the framework consists mainly of low-frequency information. Therefore, utilizing the Transformer structure, which exhibits greater sensitivity to low-frequency information, is more advantageous for facial feature extraction. To enhance the restoration of facial images, it is crucial to effectively utilize face features at local, regional, and global levels. Specifically, local regions encompass multiple pixels and are most effectively modeled using small $1 \times 1$ or $3 \times 3$ kernels, capturing typical features such as local facial details. Regional features encompass dozens of pixel points, such as eyes, nose, and other facial components. Due to their larger spatial extent, they are better modeled using convolution with large kernels [6, 27] or window-based Transformers [19]. Global features involve the structural correlation of the entire face, such as the overall facial contour, and are

**Table 2: Quantatitive evaluation for $\times 8$ FSR on CelebA [21] and Helen [13] test sets. The best and second-best results are emphasized in bold and underlined. Our method achieves the best results with the second least computational load and speed.**

| Methods | Params | FLOPs | Speed | CelebA [21] | | | | | Helen [13] | | | |
|---|---|---|---|---|---|---|---|---|---|---|---|---|
| | | | | PSNR↑ | SSIM↑ | LPIPS↓ | VIF↑ | ID↑ | PSNR↑ | SSIM↑ | LPIPS↓ | VIF↑ |
| Bicubic | - | - | - | 23.61 | 0.6779 | 0.4899 | 0.1821 | 5.9% | 22.95 | 0.6762 | 0.4912 | 0.1745 |
| FSRNet [5] | 27.5M | 40.7G | 89.8ms | 27.05 | 0.7714 | 0.2127 | 0.3852 | 66.7% | 25.45 | 0.7364 | 0.3090 | 0.3482 |
| FACN [32] | **4.4M** | 12.5G | **26.7ms** | 27.22 | 0.7802 | 0.2828 | 0.4366 | 67.1% | 25.06 | 0.7189 | 0.3113 | 0.3702 |
| DIC [23] | 22.8M | 35.5G | 120.5ms | 27.42 | 0.7840 | 0.2129 | 0.4234 | 71.6% | 26.15 | 0.7717 | 0.2158 | 0.4085 |
| SPARNet [4] | 10.6M | **7.1G** | 36.6ms | 27.73 | 0.7949 | 0.1995 | 0.4505 | 80.3% | 26.43 | 0.7839 | 0.2674 | 0.4262 |
| AD-GNN [2] | 15.8M | 15.0G | 107.9ms | 27.82 | 0.7962 | 0.1937 | 0.4470 | 81.2% | 26.57 | 0.7886 | 0.2432 | 0.4363 |
| Restormer-M [33] | 11.7M | 16.1G | 63.2ms | 27.94 | 0.8027 | 0.1933 | 0.4624 | 82.4% | 26.91 | 0.8013 | 0.2258 | 0.4595 |
| LAAT [15] | 22.4M | 8.9G | 35.1ms | 27.91 | 0.7994 | 0.1879 | 0.4624 | 84.8% | 26.89 | 0.8005 | 0.2255 | 0.4569 |
| SFMNet [28] | 8.6M | 30.6G | 49.2ms | 27.96 | 0.7996 | 0.1937 | 0.4644 | 84.6% | 26.86 | 0.7987 | 0.2322 | 0.4573 |
| **Ours** | 7.0M | 7.8G | 33.9ms | **28.04** | **0.8032** | **0.1803** | **0.4682** | **86.8%** | **27.01** | **0.8051** | **0.2148** | **0.4631** |

LR   FACN [32]   DIC [23]   SPARNet [4]   AD-GNN [2]   Restormer-M [33]   LAAT [15]   SFMNet [28]   **Ours**   GT

**Figure 5: Qualitative quality comparison for $\times 8$ FSR on CelebA [21] and Helen [13] test sets. Our method recovers more detailed face images.**

best modeled using the global Transformer. However, many methods [7, 17, 29, 33] only focus on leveraging local and global features or local and regional features. Thus, as shown in Fig. 4, we propose a full-domain Transformer as our primary module for feature extraction, which consists of two main parts: regional self-attention (RSA) focuses on extracting local and regional facial features, while global self-attention (GSA) is responsible for extracting local and global facial features. Subsequently, we will elaborate on how FDT effectively captures local, regional, and global facial features.

*Regional Self-Attention (RSA).* For an input layer normalized facial feature $X \in \mathbb{R}^{H \times W \times C}$, we first extract a set of window features from input $X$:

$$\{X_1, X_2, ..., X_n\} = Split(X), \qquad (5)$$

where $\{X_1, X_2, ..., X_n\} \in \mathbb{R}^{\frac{HW}{N^2} \times N \times N \times C}$ denotes a set of window feature patches, $N$ denotes the size of the window, and $n = \frac{H}{N} = \frac{W}{N}$. Subsequently, the model initially captures the local facial details to enhance the network's contextual information. Local details are captured by combining a $1 \times 1$ point-wise convolution and a $3 \times 3$ depth-wise convolution. Then for each window feature patch $X_i$ that enhances the local context, we project it into query $Q_i \in \mathbb{R}^{\frac{HW}{N^2} \times N^2 \times C}$, key $K_i \in \mathbb{R}^{\frac{HW}{N^2} \times N^2 \times C}$, and value $V_i \in \mathbb{R}^{\frac{HW}{N^2} \times N^2 \times C}$.

This process can be described as:

$$\{Q_i, K_i, V_i\} = \mathcal{RS}\left(\mathcal{D}\left(\mathcal{P}\left(X_i\right)\right)\right), \qquad (6)$$

where $\mathcal{RS}$ denotes a reshape operator, $\mathcal{D}$ denotes a depth-wise convolution layer, and $\mathcal{P}$ denotes a point-wise convolution layer. On this basis, for each window feature patch, regional self-attention can be formulated as:

$$\text{Attention}(Q_i, K_i, V_i) = V_i \text{ReLU}(Q_i K_i^T / \alpha), \qquad (7)$$

Here, $\alpha$ denotes a learnable parameter. To avoid the computational complexity of $O(H^2 W^2)$, we choose to implicitly encode global features across channels when computing the feature covariance $A$. Specifically, we replace the attention map of size $A \in \mathbb{R}^{HW \times HW}$ in the traditional sense with a regional attention map of size $A \in \mathbb{R}^{C \times C}$. Furthermore, to address the absence of connectivity among different windows, as shown in Fig. 4, we introduce a straightforward yet effective information exchange mechanism for RSA to facilitate communication between adjacent windows by shifting windows. Hence, our meticulous design allows RSA to enhance regional and local facial features effectively.

*Global Self-Attention (GSA).* For an input layer normalized facial feature $X' \in \mathbb{R}^{H \times W \times C}$, similarly, we initially employ $1 \times 1$ point-wise convolution and $3 \times 3$ depth-wise convolution to extract local

**Table 3: Comparison of face similarity on SCface [8] test set.**

| Methods | Average similarity↑ | | | | |
|---|---|---|---|---|---|
| | Case 1 | Case 2 | Case 3 | Case 4 | Case 5 |
| FSRNet [5] | 0.6713 | 0.6560 | 0.6794 | 0.6903 | 0.6711 |
| FACN [32] | 0.6545 | 0.6318 | 0.6571 | 0.6710 | 0.6516 |
| DIC [23] | 0.5272 | 0.4851 | 0.5772 | 0.5431 | 0.5527 |
| SPARNet [4] | 0.7100 | 0.6911 | 0.7160 | 0.7252 | 0.7041 |
| AD-GNN [2] | 0.7188 | 0.6947 | 0.7171 | 0.7283 | 0.7161 |
| LAAT [15] | 0.7193 | 0.7070 | 0.7140 | 0.7342 | 0.7238 |
| SFMNet [28] | 0.7224 | 0.7101 | 0.7243 | 0.7331 | 0.7223 |
| **Ours** | **0.7252** | **0.7239** | **0.7253** | **0.7426** | **0.7256** |

**Table 4: Ablation studies of WFD and WFU, as well as shift window (SW) mechanism and shuffle heads (SH) mechanism in the full-domain transformer on Helen [13] test set.**

| Methods | WFD | WFU | SW | SH | Params | FLOPs | PSNR / SSIM |
|---|---|---|---|---|---|---|---|
| *w/o* WFD | ✗ | ✓ | ✓ | ✓ | 0.830M | 1.131G | 26.22 / 0.7743 |
| *w/o* WFU | ✓ | ✗ | ✓ | ✓ | 0.719M | 1.085G | 26.27 / 0.7772 |
| *w/o* SW | ✓ | ✓ | ✗ | ✓ | 0.848M | 1.164G | 26.31 / 0.7763 |
| *w/o* SH | ✓ | ✓ | ✓ | ✗ | 0.848M | 1.164G | 26.31 / 0.7783 |
| **Ours** | ✓ | ✓ | ✓ | ✓ | 0.848M | 1.164G | **26.36 / 0.7795** |

information from $X'$, ensuring the accurate recovery of facial details. Subsequently, we adhere to prior methods [33] by subdividing the channel into multi-heads $h$ and concurrently learning distinct self-attention maps. Specifically, we generate query $Q \in \mathbb{R}^{h \times \hat{C} \times HW}$, key $K \in \mathbb{R}^{h \times \hat{C} \times HW}$, value $V \in \mathbb{R}^{h \times \hat{C} \times HW}$ projections based on the overall face feature after enhanced local detail, where $\hat{C}$ is the number of channels in each head. Next, we create a global attention map of size $A' \in \mathbb{R}^{\hat{C} \times \hat{C}}$ by computing the dot product of vectors $Q$ and $K$. This process emphasizes the relationships between channels while implicitly encoding global facial features. In summary, the full process of global self-attention can be formulated as follows:

$$\text{Attention}(Q, K, V) = V\text{ReLU}(QK^T/\beta), \tag{8}$$

where $\beta$ denotes a learnable parameter. To augment information exchange between the multi-heads, as illustrated in Fig. 4, we achieve this by blending multi-head feature mechanisms. Through meticulous design, our GSA effectively enhances the local and global features of the input face images.

## 4 EXPERIMENTS

### 4.1 Datasets and Evaluation Metrics

In this paper, We employ the CelebA [21] dataset for training and evaluate the models on the Helen [13] and SCface [8] datasets. Due to variations in the length and width of the original face image, we pre-detect the 68 landmarks of the face using OpenFace [1]. The face images are then cropped based on these landmarks and resized to 128×128 pixels to serve as the ground truth. The ground truth images are further downsampled to 16×16 to generate LR images using bicubic interpolation. Based on this foundation, we utilize

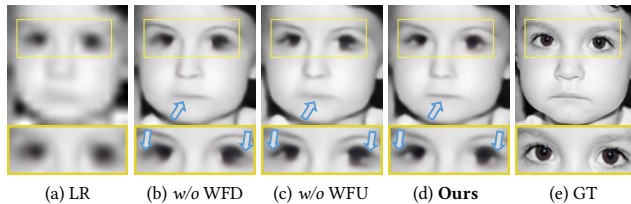

| (a) LR | (b) *w/o* WFD | (c) *w/o* WFU | (d) **Ours** | (e) GT |
|---|---|---|---|---|

**Figure 6: Impact of WFD and WFU on FSR results. We use general downsample and upsample with comparable parameters instead of WFD and WFU, respectively.**

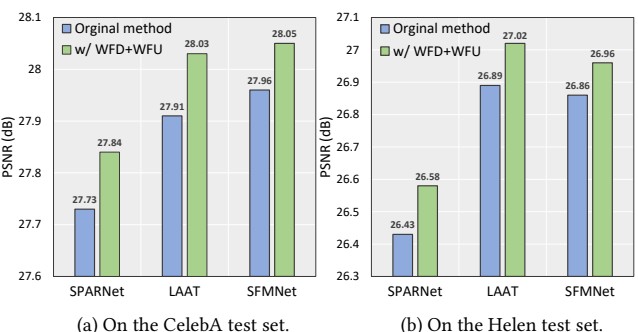

| (a) On the CelebA test set. | (b) On the Helen test set. |
|---|---|

**Figure 7: Ablation study on the generalization of WFD+WFU. We add them to methods SPARNet [4], LAAT [15], and SFM-Net [28] and observe PSNR enhancement.**

18,000 samples from CelebA for training. For testing purposes, we selected 1,000 samples from CelebA and 50 samples from Helen. As for the quality assessment metrics, we used PSNR, SSIM [30], LPIPS [34] and VIF [25]. Recognizing the significance of identity consistency, we introduced the identity comparison accuracy, denoted as ID. This metric uses SFace [35] as a recognition model, determining whether the restored and original faces belong to the same identity.

### 4.2 Implementation Details

We implement all experiments using the PyTorch framework with a single NVIDIA GeForce RTX 4090. In the network, we first extend the number of channels $C$ to 40. And in $\mathbb{R}^{H \times W \times C}$ stage, the number of full-domain Transformer is set to 2, in $\mathbb{R}^{\frac{H}{8} \times \frac{W}{8} \times 4C}$ stage the number is set to 6, and in all the remaining stages the number is set to 1. During the training stage, our model is optimized with an L1 loss with a coefficient of 1, and we use the Adam optimizer with $\beta_1$=0.9, $\beta_2$=0.99. We set the initial learning rate to $2 \times e^{-4}$ and the batchsize to 12. In addition, the ID metric's cosine threshold for identity matching is set to 0.5 in the experiment.

### 4.3 Comparisons with State-of-the-Art Methods

We benchmark our method against several state-of-the-art FSR methods using a unified dataset. The compared methods include prior-based approaches like FSRNet [5], FACN [32], and DIC [23], attention-based CNN methods such as SPARNet [4] and AD-GNN [2], and Transformer-based methods like Restormer-M [33], LAAT [15],

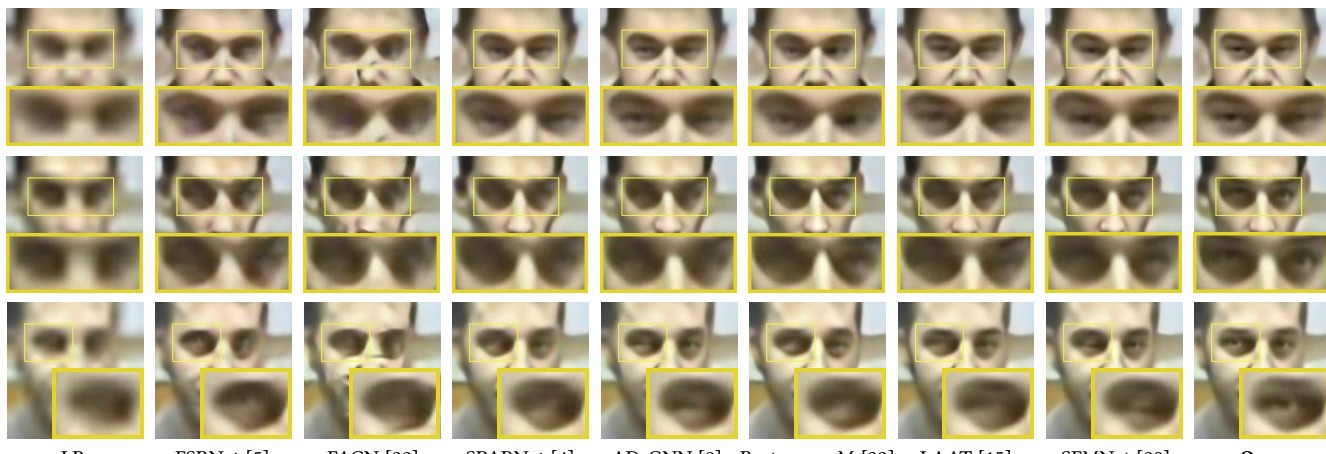

LR   FSRNet [5]   FACN [32]   SPARNet [4]   AD-GNN [2]   Restormer-M [33]   LAAT [15]   SFMNet [28]   **Ours**

**Figure 8: Qualitative comparison of state-of-the-art methods on the SCface [8] test set. Our method can restore the clearer face components, especially the eye region, which is critical for downstream face recognition tasks.**

**Table 5: Ablation studies about the efficiency of our full-domain Transformer (FDT). We use main modules in SPAR-Net [4], Restormer [33], and LAAT [15] for feature extraction instead of FDT, respectively. Our FDT can achieve gained performance with reduced computational costs.**

| Methods | Params | FLOPs | Helen [13] |
|---|---|---|---|
| | | | PSNR / SSIM / LPIPS↓ / VIF↑ |
| Ours+SPARNet [4] | 0.925M | 2.565G | 26.27 / 0.7754 / 0.2804 / 0.4241 |
| Ours+Restormer [33] | 1.063M | 1.663G | 26.33 / 0.7770 / 0.2747 / 0.4259 |
| Ours+LAAT [15] | 1.089M | 1.863G | 26.33 / 0.7771 / 0.2801 / 0.4250 |
| **Ours+FDT** | **0.848M** | **1.164G** | **26.36 / 0.7795 / 0.2745 / 0.4283** |

and SFMNet [28], where Restormer-M is a generalized image restoration method fine-tuned on face training sets. We present the quantitative results for the CelebA and Helen test datasets in TABLE 2. The best and the second-best results are emphasized in bold and underlined in this paper. Our method excels in various metrics, including image structure similarity (PSNR and SSIM), visual quality (LPIPS), fidelity (VIF), and face identity consistency (ID), achieving the best performance. Furthermore, we provide quantitative data about models, including the number of model parameters, FLOPs, and inference speed, in TABLE 2 to assess the model's efficiency. Compared with the methods above, our method is less parametric and computationally intensive, and faster in inference, exhibiting excellent efficiency. Next, we visually compare the restoration results of various methods. As shown in Fig. 5, the high-frequency face profile achieved by our method is significantly sharper and more closely resembles the ground truth, such as key facial components such as the eyes. More qualitative comparisons can also be found in *supplementary materials*.

Additionally, we validate the efficacy of our method in a practical surveillance scenario. For this purpose, we chose HR face images of test subjects from the SCface dataset as the source samples. The corresponding LR face images captured by surveillance cameras are regarded as the target samples. We created five case groups, each

consisting of 5 pairs of randomly selected face samples. The evaluation metric is the average similarity between the restored and HR faces. As shown in TABLE 3, our method consistently reconstructs faces with higher similarity in all cases, which indicates that our method can be better applied to a practical scenario. In addition, visual comparisons on the SCface test set of various methods can be found in Fig. 8, where prior-based FSR methods exhibit varying degrees of distortion in key facial components. This distortion could be attributed to inaccurate prior estimation, particularly at the current very low resolutions. Attention-based and Transformer-based methods improved the clarity of the restored face to some extent, but the face contours and edges were still not clear. In contrast, our method excels at restoring the contours of the face and facial components with superior clarity, a crucial aspect for downstream tasks like face matching. In summary, the comprehensive results, both quantitative and qualitative, illustrate the efficiency of our model's performance as well as its applicability across various scenarios.

## 4.4 Ablation Study

This subsection presents an experimental ablation analysis of the causes of our method's effectiveness, including two reasons: the proposed wavelet-based downsample and upgrade modules and the proposed full-domain Transformer.

*Wavelet Feature Downsample and Upgrade.* WFD and WFU are important components in our wavelet-based encoder-decoder structure. To assess the efficacy of our proposed WFD and WFU, we conducted experiments by substituting WFD with stride convolution for downsampling and WFU with interpolation for upsampling. As indicated in TABLE 4, the computational burden imposed by the WFD module for downsampling is nearly negligible. However, leveraging the WFD for downsampling significantly enhances the model's performance, resulting in a noteworthy PSNR gain of 0.14 dB. Subsequently, we observe that employing WFU to enhance facial details in the decoder stage yields a modest performance improvement compared to the conventional interpolation method of upsampling. This enhancement leads to a PSNR gain of 0.09 dB

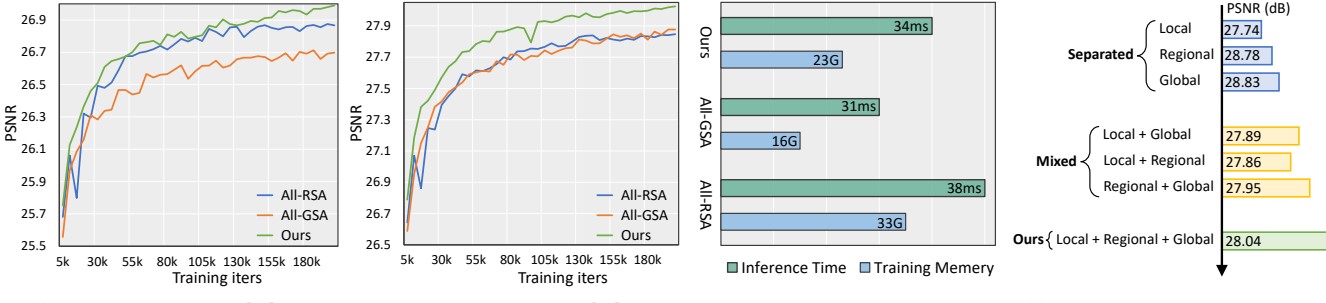

(a) Training PSNR on Helen [13] test set.    (b) Training PSNR on CelebA [21] test set.    (c) Comparison of computational loads.    (d) A comprehensive PSNR comparison.

**Figure 9: Ablation studies on the effectiveness of local, regional, and global facial features for FSR.**

while maintaining a relatively modest computational load. Meanwhile, corresponding visual comparisons are presented in Fig. 6. In Fig. 6 (b), without using WFD, the contours around the eyes, mouth, and corners of the face appear somewhat blurred. Similarly, in Fig. 6 (c), without using WFU, the contours around the right eye corner and mouth appear blurred. In contrast, Fig. 6 (d) with the complete WFD and WFU, reconstructed face component contours are sharper and closer to ground truth. This portion of the experiment strongly demonstrates the effectiveness of our WFD and WFU.

Subsequently, to assess the generalization of WFD plus WFU, we incorporate both into several existing methods, replacing their native downsampling and upsampling while preserving their proposed feature extraction modules. These methods include SPAR-Net [4], LAAT [15] and SFMNet [28]. As depicted in Fig. 7, all these methods exhibit significant performance enhancements when integrated with WFD plus WFU, with PSNR gains greater than 0.1dB. Notably, our proposed WFD and WFU are remarkably lightweight, imposing an almost negligible additional computational load. In summary of the two-part ablation studies presented above, WFD and WFU are efficient downsampling and upsampling approaches that can be seamlessly integrated into existing methods.

*Full-domain Transformer.* To assess the impact of extracting local, regional, and global facial features on facial reconstruction, we replace the combinations of RSA and GSA in FDT with all-RSA or all-GSA, respectively. This simulates existing FSR methods that exclusively focus on global and local or only regional and local facial features. Meanwhile, it can ensure that the calculated loads of the three ablation methods are comparable for a fair comparison. As depicted in Fig. 9 (a) and (b), our full-domain Transformer exhibits faster training convergence and superior performance on both test sets. Additionally, as illustrated in Fig. 9 (c), our proposed full-domain Transformer exhibits a balanced computational load, including inference speed and training memory. To further showcase the effectiveness of simultaneously capturing local, regional, and global features of a face image, we illustrate the separated case, mixed case, and our full-domain case in Fig. 9 (d). The separated case refers to situations where only one of the local, regional, or global features of the face image is focused on. Mixed case refers to situations where two of the three facial features-local, regional, and global-are attended to. In contrast to the above two cases, our method achieves a notable performance enhancement by incorporating complementary features across various scales, making it the

optimal solution. All experimental results demonstrate that simultaneously focusing on local, regional, and global features of the face image can effectively enhance performance without significantly increasing computational load.

Next, TABLE 4 substantiates the significance of information exchange mechanisms, including exchanging facial information across distinct regions via shifting windows in RSA and exchanging different multi-head information via shuffling heads in GSA. As indicated in TABLE 4, these information exchange mechanisms incur almost no computational cost, yet incorporating both mechanisms separately results in a PSNR gain of 0.05 dB in the model's performance. Moreover, we determine the efficiency of FDT by using the basic feature extraction modules in the three methods, SPARNet [4], Restormer [33], and LAAT [15], instead of FDT. As can be seen from TABLE 5, FDT achieves the best performance in several metrics with fewer numbers of parameters and FLOPs compared to these modules. This result fully demonstrates that the proposed FDT is a more efficient module to deal with FSR. Therefore, with our proposed FDT as the main feature extraction module, our method has a more powerful feature extraction capability than existing methods. More ablation studies can also be found in *supplementary materials.*

*Discussion.* Experimental results show that the efficiency of our method is contributed by two parts: wavelet-based coder-decoder structure and full-domain Transformer. Both parts can be integrated into existing encoder-decoder-based methods [4, 15, 28] to further enhance their performance. Therefore, our method is not only efficient in performance but also generalizable.

## 5 CONCLUSION

This paper presents a wavelet-based feature enhancement network for efficient FSR. To address the feature distortion caused by direct downsampling in the encoder-decoder structure, we integrate WFD and WFU into the encoder-decoder structure. Additionally, by further employing our FDT to extract low-frequency facial features comprehensively, our method can achieve a more accurate reconstruction of facial structures. We verify the effectiveness of WFD and WFU in minimizing facial structure distortion during reconstruction and the comprehensive facial feature perception capability provided by FDT. Extensive experiments, including face matching in surveillance scenarios, demonstrate that our method effectively achieves FSR with higher fidelity, achieving an excellent balance between performance, model size, and inference speed.

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
