# OpenReview forum: "Efficient Face Super-Resolution via Wavelet-based Feature Enhancement Network"
_acmmm.org/ACMMM/2024/Conference — MM2024 Poster_

### Official Review · Reviewer_WHR7 · 2024-05-22

**Rating:** 4
**Confidence:** 3

**Summary:**

To alleviate the image aliasing caused by naive downsampling operations in face super-resolution (FSR) models, this paper presents an efficient FSR method with novel wavelet-based upscaling and downscaling modules for enhanced facial details. Additionally, a domain Transformer is devised to explore and utilize latent relationships among facial features. Extensive experiments demonstrate the effectiveness and efficiency of the proposed method compared with several state-of-the-art methods.

**Strengths:**

1. Using wavelet transformations to enhance the downsampling and upsampling operations in FSR models seems interesting and effective.

2. This paper is overall well-organized and easy to follow, with clear figures to facilitate comprehension.

3. Both quantitative and qualitative results show the promising FSR performance of the proposed method.

**Limitations:**

1. Wavelet-based transformation has been widely used in super-resolution. Some classic wavelet-based works should be included for discussion and comparison, such as "Wavelet Domain Generative Adversarial Network for Multi-scale Face Hallucination" (IJCV 2019) and "Wavelet-based dual recursive network for image super-resolution" (TNNLS 2020).

2. Please revise and reorganize some sentences for grammatical correctness and logical smoothness, e.g., Lines 42-46, Lines 114-136 and Lines 144-146.

3. The underline highlighting in Table 2, columns 2-4 is not clear and needs improvement.

**Suitability:**

2

---

### Official Review · Reviewer_9DUw · 2024-05-24

**Rating:** 4
**Confidence:** 3

**Summary:**

This paper proposes a novel lightweight face super-resolution network. The main contributions include introducing a feature enhancement network based on wavelet transform, which mitigates feature distortion. Additionally, a global transformation method is proposed to enhance local, regional, and global low-frequency facial features, improving the efficiency of facial feature extraction. The proposed method achieves good evaluation metrics on widely used CelebA and Helen datasets.

**Strengths:**

The method proposed in this paper not only improves the effectiveness of face super-resolution but also balances model parameter size and runtime speed, achieving a new lightweight network for face super-resolution.

**Limitations:**

(1) The authors claim that the proposed method in this paper performs well in restoring facial contours and facial components, which are crucial for downstream tasks in facial analysis. However, the paper lacks experiments on downstream tasks. Could downstream tasks such as face recognition and face alignment be added to highlight the effectiveness of the proposed method?
(2)Some of the ablation experiments lack persuasiveness. While the authors presented visualizations of the ablation experiments for WFD and WFU in the paper, visualizations for other ablation experiments are lacking. For example, for the full-domain case proposed in this paper, specific PSNR values are shown in Figure 9(d) of the main text. Could intuitive visualizations be added to supplement the explanation?

**Suitability:**

2

---

### Official Review · Reviewer_TVmK · 2024-05-26

**Rating:** 4
**Confidence:** 3

**Summary:**

This paper describes a wavelet-based feature augmentation network for efficient FSR. To reduce feature distortion due to direct downsampling in the encoder-decoder structure, they incorporate WFD and WFU. Using our FDT to extract low-frequency face information completely leads to a more accurate reconstruction of facial structures. Extensive experiments show that their technique accomplishes FSR with high fidelity while maintaining an outstanding combination of performance, model size, and inference time.

**Strengths:**

By employing the wavelet transform to decompose the input facial features into high-frequency and low-frequency components, the method effectively addresses the common issue of feature distortion introduced by direct downsampling. This approach ensures that high-frequency features, such as edges, are preserved and processed with greater accuracy. Additionally, the integration of a full domain transformer enhances the extraction of local, regional, and global low-frequency facial features, contributing to a more comprehensive and efficient feature representation.

**Limitations:**

The method's performance has been validated through extensive experiments, but its generalizability to a wide range of face super-resolution scenarios, including diverse lighting conditions, facial expressions, and occlusions, remains to be thoroughly tested.

While quantitative evaluation shows some improvement, this marginal enhancement does not conclusively demonstrate that the results are attributable to the design module rather than the network model itself.

**Suitability:**

2

---

### Meta-Review · Area_Chair_vZ4x · 2024-06-27

**Recommendation:** Accept (Poster)
**Confidence:** 5

**Metareview:**

All 3 reviewers suggested acceptance for this paper. The AC concurs that this is a quality paper with an interesting idea of processing the high-frequency and low-frequency components from wavelet transform and with good performance demonstrated. Congratulations and pls revise the paper per the reviewer's comments.